# Effects of Underlay on Hill-Slope Surface Runoff Process of *Cupressus funebris* Endl. Plantations in Southwestern China

## Bingchen Wu and Shi Qi *

School of Soil and Water Conservation, Beijing Forestry University, Beijing 100083, China; wubingchen@bjfu.edu.cn
* Correspondence: qishi@bjfu.edu.cn; Tel.: +86-135-2204-6290

**Abstract:** Clarifying the impact of underlay (i.e., the combination of near-surface vegetation and surface micro-topography) on the surface runoff process would provide a significant theoretical basis for the adjustment of vegetation patterns and the control of soil erosion on steep slopes in mountainous areas of southwestern China. In the current study, the runoff process under different rainfall characteristics was observed based on 10 natural runoff plots, and the correlation between the spatial pattern of cypress (*Cupressus funebris*), micro-topography, and runoff characteristic parameters was tested using the Pearson correlation coefficient method. The coupling effects of the spatial pattern of cypress and micro-topography on surface runoff also were analyzed using the Response Surface Method (RSM). The results showed that (1) under the conditions of long-duration moderate rainfall or long-duration rainstorm, topographic relief, surface roughness, runoff path density, contagion index of cypress, and stand density of cypress were the main reasons for the difference in the peak flow of each runoff plot, while under the condition of the short-duration rainstorm, the factors previously mentioned were no longer the dominant factors; (2) under the conditions of long-duration heavy rainfall or long-duration rainstorm, the common laws reflected by the response of the peak flow to the composite index of the spatial pattern of cypress and micro-topography were that (1) when the composite index of the spatial pattern of cypress (V) was below 21 and the composite index of micro-topography (U) was below 10.5, the peak flow would not be significantly affected; (2) when U > 10.5, increasing the composite index of the spatial pattern of cypress within a certain range would promote peak flow; (3) when U < 7.5 and V > 18, the increase of V value could significantly reduce the peak flow, and on this basis, adjusting the V value to 41, the reduction rate of peak flow could reach 84%.

**Keywords:** underlay; spatial pattern of cypress; micro-topography; runoff process; peak flow

## 1. Introduction

Hillslope-scale surface runoff is a hydrological process that occurs on the complex underlay and is affected by multiple factors [1–3]. Vegetation, soil, and topography are the basic elements of the underlay and have an important influence on the surface runoff process. Vegetation and topography mainly affect the lateral movement of surface runoff in the underlay, while soil affects the longitudinal transmission of surface runoff through infiltration [4–6].

At present, most studies have focused on the relationship between vegetation type or quantity and runoff process [7,8]. Some studies have investigated the influence of both vegetation coverage and topographic factors on the runoff process through simulation experiments [9,10], only a few studies have pointed out that the impact of vegetation on the runoff process and hydraulic characteristics were not only related to the type and quantity of vegetation, but also to the spatial distribution of vegetation [11,12]. However, limited by the complexity of the spatial pattern and hydrological processes, which is a frontier issue in geosciences and ecology, little empirical work testing the hypothetical covariation between vegetation spatial structure and hillslope-scale surface runoff has been done [13,14].

Previous studies on the relations between vegetation pattern and hillslope runoff, often give a qualitative description of the vegetation pattern according to patch shape, distribution density, and uniformity, and then compare and analyze the difference in runoff corresponding to different spatial patterns. For example, Zhang et al. [15] indicated that a checkerboard pattern, banded pattern, and a pattern with small patches distributed like the letter X performed more effectively than a single long strip parallel to the slope direction in increasing hydraulic roughness based on artificial rainfall simulation experiments. Yang et al. [16] studied the influence of four vegetation patterns on the hydrodynamics of surface flow and found that a staggered pattern had the best effect on suppressing flow velocity.

The foregoing research showed that the impact of vegetation patterns on the runoff process was mainly reflected in the dispersion of runoff and the consumption of runoff energy. Vegetation pattern is closely related to micro-topography [17], and studies have shown that the heterogeneity of microhabitats (soil, nutrient, and water conditions in the living space of vegetation) caused by changes in micro-topography are considered to be the main factor in the development of plant species diversity and the formation of vegetation patterns [18,19]. The lateral variation of the slope, and aspect patterns, as well as the distribution of bare rock, affect the redistribution of rain, heat, and soil nutrients, indirectly defining a mosaicked pattern for vegetation assemblages [20]. On the contrary, the distribution of trees, surface vegetation, and litter caused differences in soil properties and surface roughness and changed the deposition and migration of soil particles, which indirectly reshaped the micro-topography [21].

The overlapping pattern formed by vegetation and micro-topography is the result of the long-term interaction and co-evolution of these two factors [22,23], which together determine the runoff path structure of the slope unit, and enhance or weaken the water blocking capacity of the landscape system, thereby changing the intensity and distribution of runoff [24]. Therefore, it was difficult to fully reveal the influence of complex underlay on the runoff process if the difference in the runoff generation was only attributed to the vegetation pattern without considering the overlapping pattern formed by vegetation and micro-topography. In addition to the difference in the underlay, rainfall factors affect the runoff process by affecting soil saturation and drainage network development [25]. When the rainfall exceeds a certain threshold, the fast channel of water flow was connected, resulting in the dynamic change of the surface runoff coefficient [26], thereby deepening the complexity of the impact of underlay conditions on the runoff process. Therefore, to study the effect of vegetation patterns on hillslope runoff, it was necessary to clarify the key role of vegetation pattern and micro-topography factors in the runoff process under different rainfall conditions.

The mountainous regions in southwest China have highly complex geological structures, diverse topography, and humid climates. Forest ecosystems developed in such mountainous environments have steep slopes and shallow soil characteristics, and the ecosystem was relatively vulnerable [27,28]. Cypress (*Cupressus funebris*), as the main afforestation tree species in the southwestern mountains, was widely used on steep slopes where the soil was barren and vegetation restoration was hard to achieve. This study was based on the field observation experiment of natural runoff plots. The runoff process was observed under different rainfall intensities, quantitative relations between factors of the spatial pattern of cypress and micro-topography and runoff characteristics parameters were analyzed to reveal the effect of the spatial pattern of cypress and micro-topography on the runoff process, which would provide theoretical support for the control of soil erosion on steep slopes in mountainous areas in southwestern China. In particular, because the study area was located in southwest China, where surface erosion and underground leakage loss together affected the process of rocky desertification in that region [29], while rainfall intensity [30], the development degree of underground pore fissures [31], and the strata tendencies [32] determined the ratio of surface runoff and subsurface runoff, which in turn affected the runoff process. Therefore, to eliminate the interference of subsurface runoff to this study, the ring knife samples were taken to determine the soil porosity, and

the ground penetrating radar was used to detect the underground features of the slope before the runoff plots were set up.

## 2. Materials and Methods

### 2.1. Study Sites

The study area was located in the subtropical humid monsoon climate zone, with an average annual precipitation of 1200 mm. Most of the rainfall occurred between May and August, accounting for 70% of the annual precipitation. The annual average temperature of the region was 18 °C, ranging from −2 °C to 42 °C. The study was done on a steep slope (33°) with an elevation of 565–600 m in Huaying County (30°25′21″ N, 106°50′2″ E), Sichuan Province. The soil texture belonged to limestone yellow clay, without crust, sands, or gravels. The soil bulk density of 0–15 cm (15–25 cm) under different slope position ranged from 1.31 g/cm$^3$ to 1.41 g/cm$^3$ (1.45 g/cm$^3$ to 1.54 g/cm$^3$). The soil capillary porosity of 0–15 cm (15–25 cm) under different slope position ranged from 38.54% to 42.77% (38.55% to 42.46%), the non-capillary porosity was all below 4% (Table A1), and the stable infiltration rates of different slope position ranged from 0.08 mm/min to 0.12 mm/min (Table A2), which indicated that physical properties of the soil on the slope were similar. At the same time, the detecting results (Figure A1) of the ground penetrating radar showed that the slope was a bedding slope without cracks, which indicated that there was very little leakage loss on the slope.

The slope was composed of cypress and sparse weeds (without bushes), and the litter was produced mainly by weeds and mixed together. The cypress forest on the slope originated from the Grain-for-Green Project at the beginning of the 21st century. Aerial-seeding afforestation was done on the degraded slope. After two decades of the succession of vegetation communities, the slope has developed into an open-canopied cypress forest, with significant differences in stand density and distribution patterns. The growth status of the cypress in the study area is shown in Figure 1.

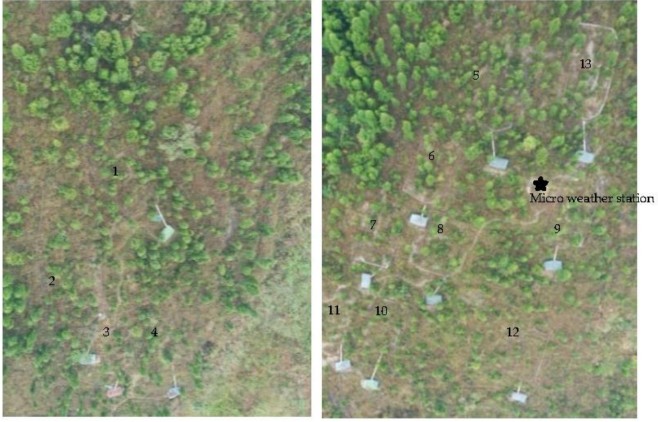

**(a)**. The layout of the runoff plot (left side of the **(b)**. The layout of the runoff plot (right side of
slope) the slope)

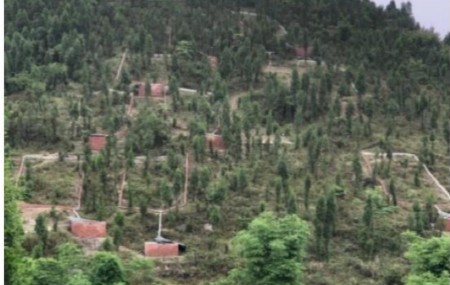

**(c)**. Overview of *Cupressus funebris* in Runoff plot

**Figure 1.** Growth status of the cypress (*Cupressus funebris*) and the layout of runoff plots in the study area. Note: (**a**,**b**) are the spatial layout of the runoff plot, and (**c**) is the panoramic view of the runoff plot.

*2.2. Research Method*

2.2.1. Runoff Plot Setting

To reflect the actual characteristics of underlays, 13 natural runoff plots (5 m × 10 m) were built from August to December 2018 (Figure 1), of which No. 11 to 13 were used as control plots in this study (the stand density of cypress in No. 11 and No. 12 runoff plots were too sparse, and the bedrock exposed rate of No. 13 was too high). The relative height difference of each runoff plot was roughly the same to ensure that the gravitational potential energy of each runoff plot was the same, so as to scientifically reflect the influence of the underlays on the hydrological process in the process of natural succession. The basic information of runoff plots 1–10 is shown in Table A3. The average crown width of cypress in each runoff plot ranged from 0.8 to 1.1 m, the average height of cypress ranged from 2.12 m to 2.91 m, and the average diameter at breast height ranged from 2.32 m to 3.15 m. The vegetation coverage (weeds other than cypress, obtained by UAV aerial photography combined with ERDAS image processing) ranged from 52.0% to 59.2%, the soil thickness ranged from 21.8 cm to 25.3 cm, the relative height difference ranged from 8.03 m to 8.44 m, and the bedrock exposed rate ranged from 0 to 0.93%. The single-factor ANOVA test of the above indicators (Table A4) among the various runoff plots showed that these indicators were not significantly different among the runoff plots. It can be considered that under the same rainfall conditions, the water holding capacity of weeds (litter of weeds) and soil in each runoff community was the same, so these factors can be regarded as irrelevant variables in this study. The tree height, the diameter at breast height, and the canopy of the cypress were all small so that the water consumption of the tree was small, and the canopy could not cover the surface, resulting in the extremely low proportion of stem flow and canopy interception accounting for the total rainfall, (0.01% and 0.1% respectively), so their impacts on hydrological process was negligible.

There was a reservoir (1 m × 2 m × 1 m) under each runoff plot with a built-in water level gauge, and the counting time interval of the water level gauge was 10 min. After each rainfall event, the water level gauge data in the reservoir was obtained, and the water and sediment in the reservoir were cleaned up through the outlet at the bottom of the reservoir. A micro weather station was set up to observe the rainfall events on the slope, and the measurement indicators included precipitation, temperature, wind direction, wind speed, solar radiation, etc.

2.2.2. Data Collection and Processing

For each runoff plot, measured data including the spatial distribution of cypress, micro-topography, rainfall, and surface runoff were collected. A Real-Time Kinematic (RTK) Global Positioning System (GPS) was used to mark the spatial position of the cypress. For the description of the spatial pattern of cypress, indicators such as the Ripley's K index [33], the contagion index [34,35], and the stand density were used for characterization. In this study, Ripley's K index described the number of individual plants in a circle with a point as the center and r was the radius, which is typically used to compare a given point distribution with a random distribution, K(r) was calculated by Equation (1).

$$
\mathrm{K(r)} = \frac{1}{N^2}\sum_{i=1}^{N}\sum_{j=1}^{N} I_r\left(u_{ij}\right) \text{with}
$$
$$
I_r\left(u_{ij}\right) = \begin{cases} 1, & u_{ij} < r \\ 0, & u_{ij} > r \end{cases} \quad \text{and } 0 \leq I_r\left(u_{ij}\right) \leq 1 \tag{1}
$$

where $N$ is the total number of trees, and $u_{ij}$ is the distance between $i$ and $j$.

The K-function can be normalized as L-function proposed by Besag [36], L(r) was calculated by Equation(2).

$$
L(r) = \sqrt{\frac{K(r)}{\pi}} - r \tag{2}
$$

A positive value of *L(r)* indicates clustering over that spatial scale whereas a negative value indicates dispersion.

The contagion index $W_i$ describes the degree of regularity of the spatial distribution of the four trees nearest to a reference tree i. $W_i$ was based on the classification of the angles between these four neighbors. A reference quantity is the standard angle $\alpha_0$, which was expected in a regular point distribution. The binary random variable $z_{ij}$ was determined by comparing each $\alpha_j$ with the standard angle $\alpha_0 = 90°$, and the contagion index $W_i$ is then defined as the proportion of angles $\alpha_j$ between the four neighboring trees which were smaller than the standard angle $\alpha_0$. $W_i$ was calculated by Equation (3).

$$W_i = \frac{1}{4} \sum_{j=1}^{4} Z_{ij} \text{with}$$
$$z_{ij} = \begin{cases} 1, & \alpha_j < \alpha_0 \\ 0, & \text{otherwise} \end{cases} \quad \text{and } 0 \leq I_r(u_{ij}) \leq 1 \tag{3}$$

$W_i$ equal to zero indicates that the trees in the vicinity of the reference tree are positioned in a regular manner, whereas $W_i$ equal to one points to an irregular or clumped distribution.

The value range and meaning of the contagion index $W_i$ are further clarified in Figure A2. In this study, the average of the contagion index, $\overline{W}$ calculated for each standard tree was used as the comprehensive contagion index of each runoff plot, and $\overline{W}$ was calculated by Equation (4).

$$\overline{W} = \frac{1}{N} \sum_{i=1}^{N} W_i \tag{4}$$

RTK-GPS also was used to measure the micro-topography for each runoff plot. During the measurement process, spatial point data were measured at 0.2 m intervals, and when encountering areas with large terrain variability, intensive measurements were done at 0.1 m intervals. Topographic relief [37], surface roughness [38], surface cutting depth [39], and runoff path density [40] were used to describe the characteristics of micro-topography for each runoff plot.

Topographic relief was calculated based on change-point theory. First, the average value $\overline{X}$ was calculated according to the elevation value of 0.2 m grid points $\{X_1, X_2, X_3 \ldots X_n\}$ in the runoff plot, and then the average topographic relief of the runoff plot could be calculated by Equation (5)

$$S = \frac{\sum_{i=1}^{n}(X_i - \overline{X})^2}{n} \tag{5}$$

where $S$ is the average topographic relief and $n$ is the number of grid points.

Surface roughness was calculated by the ratio of the surface area and the vertical projection plane of the runoff plot which were extracted using the three-dimensional (3 D) Analyst tool in ArcGIS, and it was calculated by Equation (6).

$$R = \frac{S_1}{S_2} \tag{6}$$

where $S_1$ is the surface area of the runoff plot, and $S_2$ is the vertical projection plane of the runoff plot.

The runoff path refers to the shallow trench formed by surface runoff, while the runoff path density was the total length of the runoff path per unit area. In this study, the RTK-GPS was used to measure the runoff path length, and the hydrological analysis tool in ArcGIS was used for secondary inspection. Runoff path density was calculated by Equation (7).

$$D = \frac{\sum_{i=1}^{n} L_i}{A} \tag{7}$$

where $L_i$ is the length of the *i*-th groove in the runoff plot, and $A$ is the area of the runoff plot.

The surface cutting depth refers to the difference between the average elevation and the minimum elevation of a certain point on the ground. In this study, the surface cutting

depth was calculated using the elevation data of each point in the neighborhood of the runoff path. Surface cutting depth was calculated by Equation (8).

$$H = \frac{\sum_{i=1}^{m} \left( \overline{Y_i} - Y_{imin} \right)}{m} \tag{8}$$

where $\overline{Y_i}$ is the average elevation within the neighborhood of the *i*-th point on the bottom line of the runoff path, $Y_{imin}$ is the minimum elevation of the *i*-th point on the bottom line of the runoff path, and m is the number of points on the bottom line of all runoff paths in the runoff plot.

Rainfall events and main characteristics are shown in Table A5. From June 2019 to October 2019, a total of 20 rainfall events were monitored. Among them, 8 rainfall events appeared runoff data in each runoff plot. The average rainfall intensity ranged from 2.0 to 30.0 mm/h, the rainfall duration ranges from 0.8 to 22.2 h, and the maximum 1 h rainfall intensity ranges from 5.2 to 52.2 mm/h. Among the above events, on 19 July 2019 and 22 July 2019, the runoff process of each runoff plot had no obvious peak flow (the maximum instantaneous flow during a runoff process, with obvious wave crest in the runoff process line) as the total amount of rainfall was relatively small. In the remaining 6 events, on 6, 8, and 9 August 2019, affected by the previous rainfall events, there was no significant difference in the runoff process of each runoff plot. Therefore, three rainfall events on 9 June, 28 June, and 4 August 2019 were finally screened out, representing three types of rainfall events including long-duration moderate rainfall (the rainfall lasted more than 3 h, and the average rainfall intensity was between 1.5–2.5 mm/h), long-duration rainstorm (the rainfall lasted more than 3 h, and the average rainfall intensity exceeded 2.5 mm/h), and short-duration rainstorm (the rainfall lasted no more than 3 h, and the maximum 1 h rainfall intensity exceeded 30 mm/h).

### 2.3. Statistical Methods

To clarify the key factors that caused the difference in the runoff process of each runoff plot, the Pearson correlation coefficient method was used to test the correlation between the factors of the spatial patterns of cypress (stand density of cypress, L(r) index of cypress, and contagion index of cypress) and micro-topography (topographic relief, surface roughness, surface cutting depth, and runoff path density), and the characteristic parameters (peak flow, runoff duration) of the runoff process. According to the effecting direction of each factor on the runoff process, the composite index of micro-topography and the spatial pattern of cypress are constructed. For example, when topographic relief and runoff path density are significantly positively correlated with the characteristic parameter of the runoff process, but surface roughness is significantly negatively correlated with the parameter, then topographic relief*runoff path density/surface roughness is used as the composite index of micro-topography. Similarly, the composite index of the spatial pattern of cypress can be obtained. The purpose of constructing the composite index of micro-topography and the spatial pattern of cypress was to quantify the overall characteristics of the micro-topography and the spatial pattern of cypress, and then reflect the coupling influence of the micro-topography and the spatial pattern of cypress on runoff process. Since the factors that made up the micro-topography or the spatial pattern of cypress were independent of each other and had different magnitudes, according to the correlation coefficient of each factor with the characteristic parameters of the runoff process, the influence of each factor on the runoff process can be included in the quantitative expression of the composite index by multiplying the values in the same direction and dividing the other values in the opposite direction.

Taking these 10 runoff plots as samples, the Response Surface Method (RSM) was used to construct a three-dimensional surface equation between the composite index of

micro-topography and the spatial pattern of cypress, and the characteristic parameters of the runoff process. The response surface equation was shown in Equation (9).

$$Q_p = aU^2 + bV^2 + cU \times V + dU + eV + f \tag{9}$$

where Qp is the peak flow (or other characteristic parameters of the runoff process), U is the composite index of micro-topography, V is the composite index of the spatial pattern of cypress, and a, b, c, d, e, f are fitting parameters.

## 3. Results

### 3.1. Typical Rainfall Events and Runoff Process in Runoff Plots

Typical rainfall events of long-duration moderate rainfall, long-duration rainstorm, short-duration rainstorm, and the corresponding runoff process lines of each runoff plot were shown in Figures 2–4. Comparing the runoff process under three typical rainfall events (Figures 2–4), it can be seen that when the rainfall event was the same, the peak flow in each runoff plot was different, but the time to reach the peak was the same. The time of each runoff plot to reach the peak showed long-duration moderate rainfall (140 min) > long-duration rainstorm (100 min) > short-duration rainstorm (70 min).

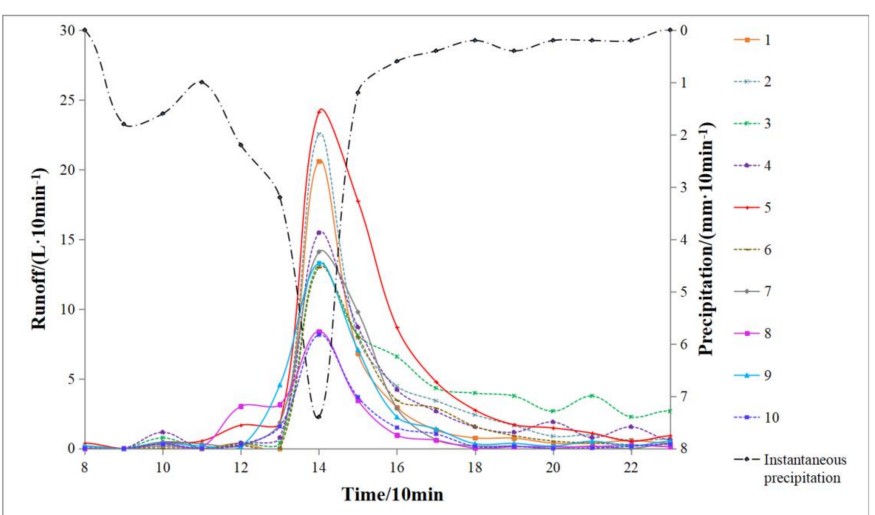

**Figure 2.** Surface runoff process under long-duration moderate rainfall on 9 June 2019.

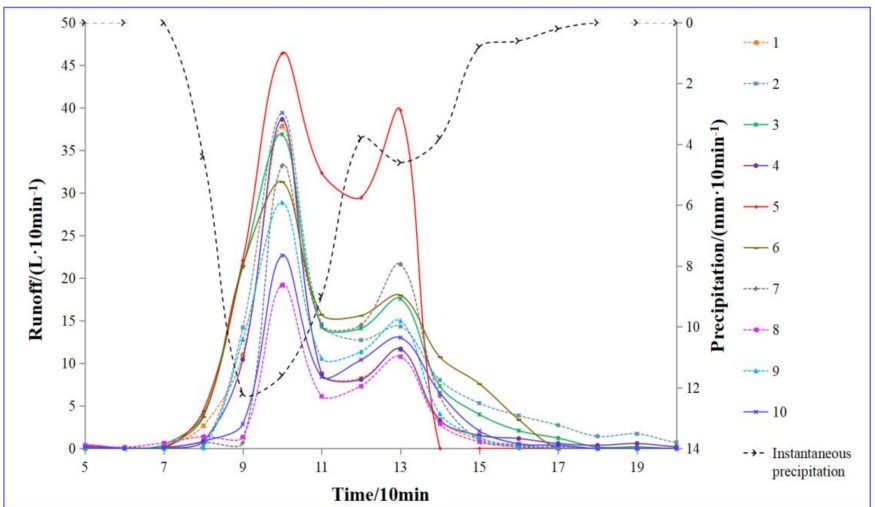

**Figure 3.** Surface runoff process under long-duration rainstorm on 28 June 2019.

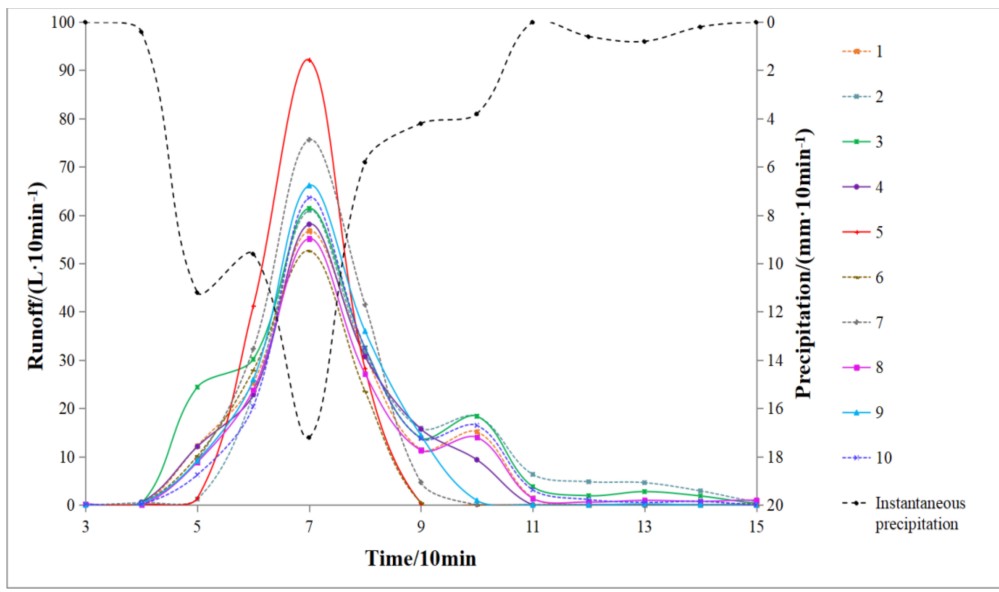

**Figure 4.** Surface runoff process under short-duration rainstorm on 4 August 2019.

The peak flow coefficient of each runoff plot under three typical rainfall events was shown in Table 1. From the table, the peak flow coefficient of all runoff plots showed short-duration rainstorm (0.293–0.514) > long-duration rainstorm (0.117–0.282) > long-duration moderate rainfall (0.087–0.257), indicating that with the increase of rainfall intensity and the concentration of precipitation, the blocking effect of different underlays on surface runoff decreased.

**Table 1.** Differences in peak flow coefficient of each runoff plot under different rainfall conditions.

| Rainfall Characteristics | Peak Flow Coefficient | | | | | | | | | |
|---|---|---|---|---|---|---|---|---|---|---|
| | 1 | 2 | 3 | 4 | 5 | 6 | 7 | 8 | 9 | 10 |
| Long-duration moderate rainfall | 0.219 | 0.240 | 0.141 | 0.164 | 0.257 | 0.138 | 0.150 | 0.089 | 0.141 | 0.087 |
| Long-duration rainstorm | 0.230 | 0.239 | 0.224 | 0.235 | 0.282 | 0.190 | 0.202 | 0.117 | 0.175 | 0.138 |
| Short-duration rainstorm | 0.316 | 0.340 | 0.342 | 0.324 | 0.514 | 0.293 | 0.422 | 0.307 | 0.369 | 0.354 |

Note: The peak flow coefficient is the total flow divided by total rainfall at the time of reaching the peak.

### 3.2. Impact of the Spatial Pattern of Cypress/Micro-Topography on Peak Flow

The correlation among the factors of the spatial patterns of cypress and micro-topography and peak flow under three typical rainfall events was shown in Table 2. For peak flow, under the condition of long-duration moderate rainfall or long-duration rainstorm, topographic relief, surface roughness, runoff path density, contagion index of cypress, and stand density of cypress were significantly correlated with peak flow ($p < 0.05$), indicating that these five factors were the main reasons for the difference in the peak flow in each runoff plot under these two conditions. Among them, topographic relief, runoff path density were significantly positively correlated with peak flow, and surface roughness, contagion index of cypress, stand density of cypress were significantly negatively correlated with peak flow. However, under the condition of a short-duration rainstorm, no significant correlation was found between each factor and the peak flow, indicating that the underlay was no longer the dominant factor affecting the peak flow.

**Table 2.** Pearson's coefficients of bivariate correlations between the characteristic parameters of the spatial pattern of cypress (*Cupressus funebris*) and micro-topography and peak flow.

| | Topographic Relief | Surface Rough-ness | Surface Cutting Depth | Runoff Path Density | L(d) Index of Cypress | Contagion Index of Cypress | Stand Density of Cypress | Peak Flow (Long-Duration Moderate Rainfall) | Peak Flow (Long-Duration Rainstorm) |
|---|---|---|---|---|---|---|---|---|---|
| Surface roughness | −0.328 | | | | | | | | |
| Surface cutting depth | 0.132 | −0.513 | | | | | | | |
| Runoff path density | 0.101 | −0.645 * | 0.496 | | | | | | |
| L(d) index of cypress | 0.186 | 0.052 | 0.455 | −0.345 | | | | | |
| Contagion index of cypress | −0.737 * | 0.228 | 0.169 | −0.316 | 0.170 | | | | |
| Stand density of cypress | −0.358 | 0.547 * | −0.202 | −0.486 | −0.208 | 0.559 | | | |
| Peak flow (Long-duration moderate rainfall) | 0.685 * | −0.744 * | 0.571 | 0.736 * | 0.080 | −0.647 * | −0.691 * | | |
| Peak flow (Long-duration rainstorm) | 0.693 * | −0.656 * | 0.237 | 0.689 * | −0.157 | −0.749 * | −0.717 * | 0.898 ** | |
| Peak flow (Short-duration rainstorm) | 0.760 * | −0.073 | 0.275 | −0.025 | 0.389 | −0.421 | −0.161 | 0.455 | 0.481 |

* denotes significant differences at $p < 0.05$ level, and ** denotes significant differences at $p < 0.01$.

*3.3. Coupling Effects of the Spatial Pattern of Cypress and Micro-Topography on Peak Flow under the Condition of Long-Duration Moderate Rainfall/Long-Duration Rainstorm*

3.3.1. Correlation between the Composite Index of the Spatial Pattern Cypress/Micro-Topography and Peak Flow

Based on the significance and correlation coefficient of the impact of each influencing factor on the peak flow, the composite index of micro-topography (topographic relief × runoff path density/surface roughness) and the composite index of the spatial pattern of cypress (contagion index of cypress × stand density of cypress) were obtained.

The correlation between the composite index of micro-topography and the composite index of the spatial pattern of cypress and the peak flow was shown in Table A6. The results showed that under the long-duration moderate rainfall or long-duration rainstorm, the composite index of micro-topography was significantly positively correlated with the peak flow, and the composite index of the spatial pattern of cypress was significantly negatively correlated with the peak flow.

3.3.2. Coupling Effects of the Spatial Pattern of Cypress and Micro-Topography on Peak Flow under the Condition of Long-Duration Moderate Rainfall/Long-Duration Rainstorm

The response surface equation of peak flow to the composite indexes of the spatial pattern of cypress and micro-topography under the condition of long-duration moderate rainfall/long-duration rainstorm were constructed using the RSM which were shown in Equations (10) and (11):

$$Q_{p1} = 0.316U_1{}^2 - 0.02V_2{}^2 + 0.079U_1 \times V_1 - 5.51U_1 + 0.29V_1 + 28.5 \tag{10}$$

$$Q_{p2} = 0.517U_2{}^2 - 0.0396V_2{}^2 + 0.35U_2 \times V_2 - 13.8U_2 - 1.04V_2 + 95 \tag{11}$$

Figure 5 showed the response surface of the two conditions. According to the change trend of the peak flow with the composite index of the spatial pattern of cypress or micro-topography, the response surface was divided into 4 regions (I, II, III, IV) in the two conditions. The value range of five factors and two composite indexes in each region are shown in Table A7.

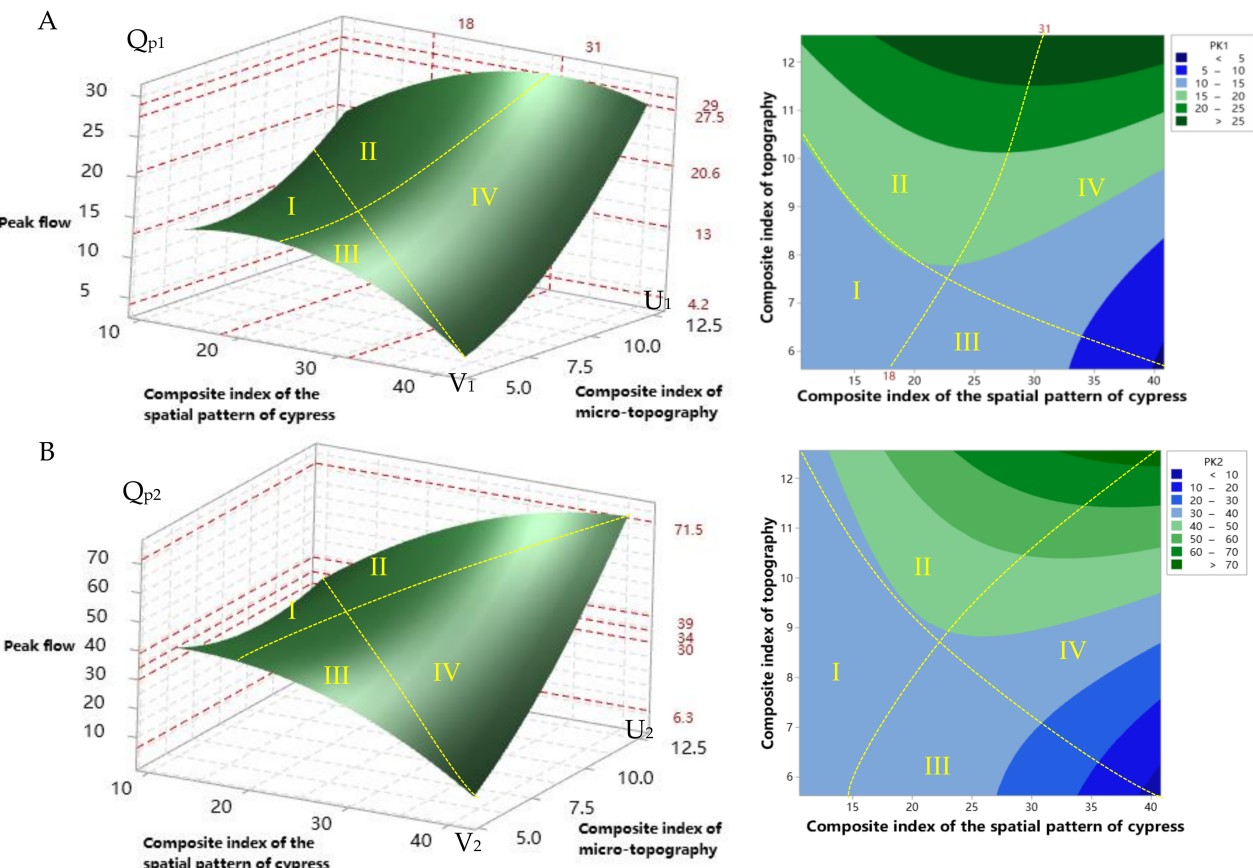

**Figure 5.** Response surface of the peak flow to the coupling of the composite index of the spatial pattern of cypress (*Cupressus funebris*) and the composite index of micro-topography (note: (**A**) was the response surface under long-duration moderate rainfall, and (**B**) was the response surface under long-duration rainstorm. According to the partial derivative analysis of the response surface, the surface was divided into 4 regions by dashed lines, and each region presented a different trend of change. Regions I, II, III, and IV respectively showed: $Q_p$ did not change significantly with the changes of U and V; $Q_p$ slightly increased with the increase of V; $Q_p$ decreased significantly with the increase of V; $Q_p$ increased significantly with the increase of U. PK1 means peak flow under long-duration moderate rainfall, PK2 means peak flow under long-duration rainstorm).

Under the condition of long-duration moderate rainfall/long-duration rainstorm, the common laws of the coupling effects of the composite index of the spatial pattern of cypress (V) and the composite index of micro-topography (U) on peak flow were: (1) When V < 21, U < 10.5 (I area), the peak flow in this area did not significantly change with the changes of the composite indexes of the spatial pattern of cypress or micro-topography. The main feature of this underlay was that the surface roughness was between 1.65 to 1.72 (10%–15% higher than the average value of the 10 runoff plot), the runoff path density was between 7.9 to 8.02 (9%–11% lower than the average value), the stand of cypress was between 34 to 36 (20%–24% lower than the average value), the contagion index of cypress was between 0.4 to 0.43 (14%–20% lower than the average value). (2) When the U value reached the condition to significantly increase the peak flow (U > 10.5, corresponding to the areas of $Q_{p1}$ > 20 and $Q_{p2}$ > 50 in Figure 5), increasing the V value within a certain range would not reduce but increase the peak flow. The main feature of this underlay was that the surface roughness was between 1.21 to 1.35 (9%–11% lower than the average value), the runoff path density was between 9.62 to 11.22 (8%–26% higher than the average value), and the stand density of cypress was between 22 to 36 (20%–51% lower than the average value). (3) When U < 7.5, V > 18 (area III in Figure 5), increasing the V value significantly reduced the peak flow. The main feature of this underlay was that the runoff path density was between 6.96 to 7.52 (16%–22% lower than the average value), the stand density of cypress was between

54 to 70 (20%–56% higher than the average value), this area was regarded as the main area for adjusting the spatial pattern of cypress to control peak flow. (4) When the composite index of the spatial pattern of cypress exceeded a certain value (the dividing line of the change from area I, II to III, IV, the value increased with the increase of U value), as the U value increased, the dominant factor affecting the peak flow changed from the spatial pattern of cypress to micro-topography.

In these two conditions, the main difference in response surface was the magnitude of the change of the peak flow with the change of V value or U value. For example, when the V value took the maximum value (V = 41), the peak flow under the condition of long-duration moderate rainfall increased from 4.2 to 27.5 (an increase of 554.7%) with the increase of U value, while the peak flow under the condition of long-duration rainstorm increased from 6.3 to 71.5 (an increase of 1034.9%) with the increase of U value. When the U value took the minimum value (U = 5.5), the peak flow under the condition of long-duration moderate rainfall decreases from 13 to 4.2 (a decrease of 67.7%) with the increase of V value, while the peak flow under the condition of long-duration rainstorm decreased from 34 to 6.3 (a decrease of 81.5%) with the increase of V value.

## 4. Discussion

### 4.1. Interaction between the Spatial Pattern of Cypress and Micro-Topography

According to the correlation test between the factors of the spatial pattern of cypress and micro-topography (Table 2), the interaction between the spatial pattern of cypress and micro-topography was mainly reflected in the negative correlation between the contagion index of cypress and the topographic relief (the correlation coefficient was −0.737), and the positive correlation between the stand density of cypress and surface roughness (the correlation coefficient was 0.547).

Generally speaking, the surface roughness was often affected by surface vegetation, litter, gravel, rocks, soil particle composition, and rainfall-runoff process, and the main reason was that the rainfall-runoff process drove the movement of soil particles [41], while vegetation, rocks and other obstacles affected the sorting process of the soil particles, which gradually caused changes in the roughness of the soil surface [42–46]. However, in this study area, the high soil viscosity prevented the difference in surface height from changing significantly in the short-term rainfall-runoff process, and there was no significant difference in the coverage of surface weeds, soil thickness, and soil physical properties in each runoff plot. For the interaction between the stand density of cypress and surface roughness, the main consideration was the shaping effect of individual trees on micro-topography [47] and the influence of surface roughness on soil infiltration and soil nutrients [48]. The increase in the stand density of cypress made more micro-habitats form on the slope and caused the difference in surface roughness in each micro-habitat, which finally resulted in a positive feedback mechanism between the stand density of cypress and surface roughness.

For the interaction between the contagion index of cypress and topographic relief, related studies have shown that the formation of the spatial pattern of vegetation strengthens the source-sink effect on the migration of soil material [49]. The resistance of the vegetation patch increased the flow velocity along the edge of the patch, and the formation of the plume structure inhibited soil erosion above the patch, while it enhanced the soil erosion below the patch [50–52]. The continuous spatial migration of soil particles made the otherwise uniform slope become undulating. In this study, the higher contagion index of cypress (greater than 0.5) indicated that the spatial pattern of cypress was in a clumped distribution, which was more unfavorable for runoff to pass than a regular distribution. These clumped distributions had a stabilizing effect on the soil in the micro-habitat, reducing the topographical fluctuations caused by soil migration.

On the other hand, convexity and elevation were the most important variables affecting the distribution of trees [53]. The impact of convexity on the distribution of trees mainly came from the redistribution of soil and water which led to the spatial heterogeneity of micro-habitats, while the clumped distribution of trees in the karst mountainous area

was closely related to the high heterogeneity of micro-habitats and the restriction of seed dispersal [54]. The increase in topographic relief reduces the maximum gathering radius of trees [55], which indicates that higher topographic relief is not conducive to the formation of a clumped distribution of cypress (low contagion index of cypress).

### 4.2. Differences in the Impact of Different Spatial Pattern of Cypress, Micro-Topography and Rainfall Conditions on Peak Flow

The peak flow was an important parameter to express the intensity of soil erosion or water loss on steep slopes [56]. In this study, the results in Table 1 indicated that with the increase of rainfall intensity and the concentration of precipitation, the blocking effect of different underlays on surface runoff decreased. Studies have shown that the micro-topography mainly affects the runoff velocity during the runoff process, which was specifically reflected in the surface resistance provided by the surface roughness, the confluence channel provided by the runoff path, and the change of runoff energy caused by topography relief [57,58]. As a result, lower surface roughness, higher topographic relief, and runoff path density made the underlay less resistant, which was more conducive to the connection of the drainage network of the slope, which could explain the law that when the U value reached the condition to significantly increase the peak flow (U > 10.5, corresponding to the areas of $Q_{p1} > 20$ and $Q_{p2} > 50$ in Figure 5), increasing the V value within a certain range would not reduce but increase the peak flow. In this condition, the micro-topography had already played a decisive role in the impact of the peak flow, and it was more conducive to the formation of a mechanism for promoting runoff between the cypress and its local micro-topography (the plume on both sides of the cypress promotes the formation of shallow trenches). With the increase of rainfall intensity, on the one hand, the preparation time for the consumption of runoff energy on the rough surface is reduced and the time for water to reach the runoff path is shortened [59,60], thereby accelerating the self-organization process of confluence network on the slope [61], which was the main reason why the runoff time to reach the peak flow under high rainfall intensity was significantly less than that of low rainfall intensity.

On the other hand, under steep slope conditions, the downstream-moving force produced by rainfall was greater than the upstream-moving force, which could reduce the resistance of the surface flow and increase the flow velocity on the slope, and the decreased effect of resistance would increase with an increase in rainfall intensity [62,63]. Therefore, the increase in rainfall intensity and the concentration of precipitation increased the promotion effect of micro-topography on the peak flow. Among the different runoff plots in Table 1, the increase in the peak flow coefficient varied among the plots with the increase of rainfall intensity. The main reason was that under different combinations of micro-topography and the spatial pattern of cypress, the dominant factors affecting the peak flow also were different.

### 4.3. Strategies for the Adjustment of Vegetation Pattern on Slopes in Southwest Mountainous Areas

Although the underlay features may be the product of the hydrological process of the slope, the combination of land preparation techniques and vegetation could be an effective way to combat soil degradation on vulnerable, steep slopes [62,63]. The topography and geomorphology conditions of the mountainous areas in southwestern China are relatively harsh, and the economic conditions of the mountainous areas are difficult to achieve large-scale topographic reconstruction to prevent soil erosion. By clarifying the interaction between vegetation pattern and micro-topography, as well as the coupling effect of vegetation pattern and micro-topography on surface runoff, the current soil erosion can be improved by adjusting the vegetation pattern to promote the suppression mechanism of the underlay on surface runoff.

The results of the response of the peak flow to the composite index of the spatial pattern of cypress/micro-topography under the condition of long-duration moderate rainfall/long-duration rainstorm showed that when to reduce the peak flow, the adjustment of the vegetation pattern had certain prerequisites: When the composite index of micro-

topography was small and the composite index of the spatial pattern of cypress reached a certain value (the area III in Figure 5, the feature was that there was no obvious shallow groove on the underlay, and the stand density of cypress reached a certain level), the peak flow can be reduced by replanting cypress to form a clumped structure among trees. Under the conditions of long-duration moderate rainfall/long-duration rainstorm, when the composite index of the spatial pattern of cypress was adjusted from 18 (15) to 41, the peak flow can be reduced from 39.6 (14.2) to 6.3 (4.2), and the reduction rate reached 84% (70%). When the composite index of micro-topography reached the condition to significantly increase the peak flow (the areas II and IV in Figure 5, the feature was that the underlay had obvious shallow trenches and the topography was undulating), then the micro-topography became the dominant factor affecting the peak flow. It was easier to form a runoff promotion mechanism between cypress and local micro-topography. At this time, the focus of the adjustment of vegetation patterns should be to eliminate the promotion effect of cypress and local micro-topography on surface runoff. If obvious shallow trenches had been formed on both sides of a certain cypress, this cypress should be cut. Otherwise, it was necessary to supplement certain local micro-topography modification measures for this cypress, such as shallow trench cut-off measures or land consolidation measures.

## 5. Conclusions

The combinations of different micro-topography and the spatial pattern of cypress under different rainfall characteristics had an important impact on peak flow. Under the condition of long-duration moderate rainfall or long-duration rainstorm, among the characteristic parameters of micro-topography and the spatial pattern of cypress, topographic relief, surface roughness, runoff path density, contagion index of cypress, and stand density of cypress were the main reasons for the difference in the peak flow of each runoff plot, while under the condition of the short-duration rainstorm, the factors previously mentioned were no longer the dominant factors.

When using the adjustment of the spatial pattern of cypress to reduce the peak flow, it should be determined by the characteristics of the underlay. The key point was to suppress (promote) the positive (negative) feedback of runoff generation between cypress and local micro-topography. Specifically, under the conditions of long-duration moderate rainfall or long-duration rainstorm, when $V > 18$ and $U < 7.5$, adjusting the V value to 41, and the reduction rate of peak flow could reach 84%.

With the increase of rainfall intensity and the concentration of precipitation, the blocking effect of different underlays on surface runoff decreased. This was specifically reflected in the changes in the degree of impact of the factors of micro-topography and the spatial pattern of cypress on the peak flow. Subsequent studies should be done on the rainfall intensity thresholds corresponding to the effect of each factor (especially surface roughness, because the increase of the splashing potential energy of raindrops may cause the change of the surface roughness) on the peak flow, to propose an optimization for the underlay conditions to achieve the purpose of efficiently controlling soil erosion on a slope.

**Author Contributions:** Conceptualization, S.Q.; methodology, B.W.; validation, B.W.; formal analysis, B.W.; investigation, B.W.; writing—original draft preparation, B.W.; writing—review and editing, B.W.; visualization, B.W.; supervision S.Q.; funding acquisition S.Q. All authors have read and agreed to the published version of the manuscript.

**Funding:** This research was funded by the National Key Research and Development Program of China (No. 2017YFC0505602).

**Informed Consent Statement:** Informed consent was obtained from all subjects involved in the study.

**Data Availability Statement:** Data will be made available on request.

**Acknowledgments:** Much appreciation to the Forestry Bureau of Huaying City, Sichuan Province for supporting this study.

**Conflicts of Interest:** The authors declare no conflict of interest.

## Appendix A

**Table A1.** Soil physical properties of different slope positions.

| No | Soil Depth/cm | Soil Bulk Density/(g/cm³) | Capillary Porosity/% | Non-Capillary Porosity/% | Total Porosity/% |
|----|---------------|---------------------------|----------------------|--------------------------|------------------|
| I | 0–15 | 1.35 | 38.54 | 1.63 | 40.17 |
| | 15–25 | 1.49 | 42.34 | 3.28 | 45.61 |
| II | 0–15 | 1.38 | 39.86 | 2.26 | 42.12 |
| | 15–25 | 1.47 | 38.55 | 1.93 | 40.48 |
| III | 0–15 | 1.41 | 41.82 | 3.31 | 45.13 |
| | 15–25 | 1.50 | 40.69 | 2.50 | 43.20 |
| IV | 0–15 | 1.31 | 42.55 | 2.56 | 45.11 |
| | 15–25 | 1.50 | 42.46 | 2.05 | 44.51 |
| V | 0–15 | 1.31 | 42.77 | 3.71 | 46.48 |
| | 15–25 | 1.45 | 41.60 | 3.86 | 45.46 |
| VI | 0–15 | 1.36 | 41.65 | 2.69 | 44.34 |
| | 15–25 | 1.54 | 38.69 | 2.19 | 40.88 |

Note: The ring knife samples I to VI were evenly distributed in different slope positions.

## Appendix B

**Table A2.** Soil infiltration rate of different slope positions.

| Time | Infiltration Rate/(mm/min) | | | | | |
|------|-----|------|------|------|------|------|
| | I | II | III | IV | V | VI |
| 0 min–1 min | 0.25 | 0.30 | 0.45 | 0.25 | 0.20 | 0.35 |
| 1 min–2 min | 0.2 | 0.28 | 0.20 | 0.15 | 0.15 | 0.20 |
| 2 min–3 min | 0.20 | 0.25 | 0.20 | 0.15 | 0.15 | 0.15 |
| 3 min–4 min | 0.20 | 0.25 | 0.20 | 0.10 | 0.15 | 0.15 |
| 4 min–5 min | 0.20 | 0.25 | 0.15 | 0.10 | 0.18 | 0.15 |
| 5 min–6 min | 0.15 | 0.25 | 0.15 | 0.09 | 0.16 | 0.15 |
| 6 min–7 min | 0.15 | 0.25 | 0.10 | 0.09 | 0.15 | 0.10 |
| 7 min–8 min | 0.10 | 0.20 | 0.10 | 0.10 | 0.10 | 0.10 |
| 8 min–9 min | 0.10 | 0.20 | 0.08 | 0.10 | 0.10 | 0.10 |
| 9 min–10 min | 0.10 | 0.15 | 0.08 | 0.10 | 0.10 | 0.10 |
| 10 min–15 min | 0.10 | 0.18 | 0.08 | 0.10 | 0.09 | 0.09 |
| 15 min–20 min | 0.11 | 0.19 | 0.10 | 0.09 | 0.08 | 0.09 |
| 20 min–25 min | 0.10 | 0.12 | 0.10 | 0.11 | 0.08 | 0.09 |
| 25 min–30 min | 0.10 | 0.12 | 0.10 | 0.08 | 0.08 | 0.09 |
| 30 min–35 min | 0.10 | 0.12 | 0.09 | 0.08 | 0.08 | 0.09 |
| 35 min–40 min | 0.10 | 0.12 | 0.10 | 0.08 | 0.08 | 0.09 |

Note: Samples I to VI were evenly distributed in different slope positions.

## Appendix C

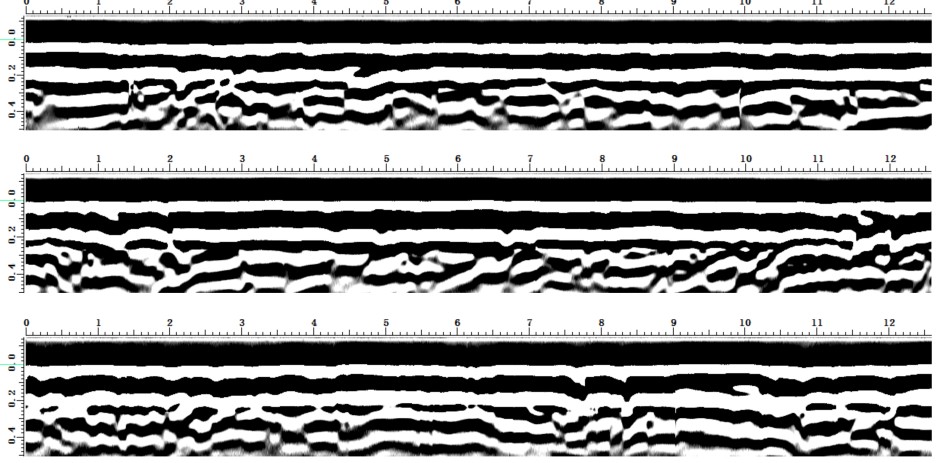

**Figure A1.** Rock characteristics of different slope positions.

## Appendix D

**Table A3.** Characteristics of each runoff plot (note: DBH means diameter at breast height).

| No | Average Crown Width of Cypress/(m × m) | Average Height of Cypress/m | Average DBH of Cypress/cm | Surface Vegetation Coverage/% | Soil Thickness/cm | The Relative Height Difference of Runoff Plots/m | Bedrock Exposure Rate/% | Topographic Relief/m | Surface Roughness | Surface Cutting Depth/cm | Runoff Path Density/ (m/100m²) | L(r) Index of Cypress | Contagion Index of Cypress | Stand Density of Cypress/ (Plants/100 m²) |
|---|---|---|---|---|---|---|---|---|---|---|---|---|---|---|
| 1 | 0.8 × 1.0 | 2.32 | 2.78 | 58.8 | 24.5 | 8.1 | 0 | 1.54 | 1.72 | 5.94 | 8.02 | 0.16 | 0.43 | 36 |
| 2 | 0.8 × 0.9 | 2.12 | 2.73 | 58.2 | 22.5 | 8.13 | 0 | 1.31 | 1.73 | 7.43 | 8.08 | 0.19 | 0.6 | 68 |
| 3 | 0.8 × 0.8 | 2.83 | 2.68 | 55 | 23.4 | 8.17 | 0.93 | 1.36 | 1.21 | 6.94 | 10.08 | −0.05 | 0.53 | 32 |
| 4 | 0.9 × 1.0 | 2.69 | 3.09 | 57.4 | 24.5 | 8.44 | 0.14 | 1.71 | 1.31 | 10.76 | 9.62 | 0.22 | 0.45 | 36 |
| 5 | 0.8 × 0.9 | 2.46 | 2.32 | 52 | 23.6 | 8.03 | 0.76 | 1.56 | 1.45 | 6.32 | 7.52 | 0.03 | 0.47 | 70 |
| 6 | 0.9 × 1.0 | 2.76 | 3.04 | 57.1 | 23.8 | 8.22 | 0.83 | 1.31 | 1.62 | 8.73 | 6.96 | 0.26 | 0.64 | 54 |
| 7 | 0.9 × 1.0 | 2.72 | 2.45 | 55.8 | 21.8 | 8.12 | 0 | 1.44 | 1.35 | 10.32 | 11.22 | 0.08 | 0.42 | 28 |
| 8 | 0.9 × 1.1 | 2.4 | 3.02 | 53.5 | 25.3 | 8.32 | 0.93 | 1.39 | 1.61 | 6.95 | 9.92 | −0.13 | 0.54 | 66 |
| 9 | 0.8 × 0.9 | 2.54 | 3.15 | 59.2 | 22 | 8.03 | 0.75 | 1.51 | 1.65 | 3.49 | 7.9 | 0.03 | 0.4 | 34 |
| 10 | 0.8 × 1.0 | 2.91 | 2.65 | 58.4 | 22.1 | 8.33 | 0.63 | 1.54 | 1.23 | 8.71 | 9.64 | 0.22 | 0.48 | 22 |
| 11 | 0.8 × 0.8 | 2.42 | 2.84 | 57.2 | 23.6 | 8.13 | 0.26 | 1.37 | 1.15 | 5.74 | 8.4 | 0.26 | 0.69 | 6 |
| 12 | 0.8 × 0.8 | 2.66 | 2.73 | 59.6 | 22.8 | 8.05 | 0 | 1.31 | 1.19 | 5.46 | 8.22 | 0.28 | 0/72 | 10 |
| 13 | 0.9 × 1.0 | 2.52 | 2.78 | 52.1 | 22.4 | 8.19 | 6.85 | 1.36 | 1.61 | 6.76 | 10.32 | 0.18 | 0.56 | 52 |

Note: The L(r) index in the table was explained in Equations (1) and (2).

## Appendix E

**Table A4.** One-way ANOVA test of irrelevant variables in runoff plots.

| Variables | Mean Square | F | Significance |
|---|---|---|---|
| Average crown width of *Cupressus funebris* | 0.007 | 0.482 | 0.637 |
| Average height of *Cupressus funebris* | 0.098 | 1.858 | 0.225 |
| Average DBH of *Cupressus funebris* | 0.094 | 1.275 | 0.337 |
| Surface vegetation coverage | 6.152 | 1.079 | 0.391 |
| Soil thickness | 2.490 | 2.114 | 0.191 |
| The relative height difference | 0.002 | 0.073 | 0.930 |
| Bedrock exposure rate | 0.072 | 0.370 | 0.703 |

## Appendix F

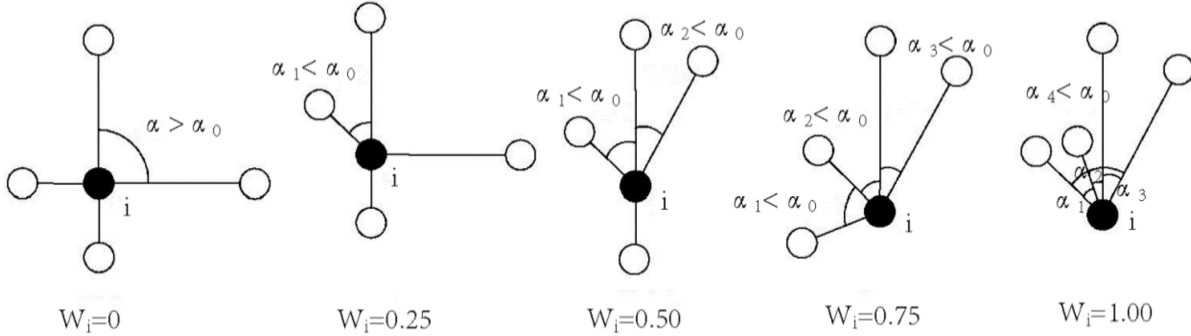

**Figure A2.** Value range and meaning of the contagion index $W_i$ ($W_i = 0$, four $\alpha$ angles all are greater than or equal to $\alpha_0$; $W_i = 0.25$, only one $\alpha$ angles is smaller than $\alpha_0$; $W_i = 0.50$, only two $\alpha$ angles are smaller than $\alpha_0$; $W_i = 0.75$, only three $\alpha$ angles are smaller than $\alpha_0$; $W_i = 1.00$, four $\alpha$ angles are all smaller than $\alpha_0$).

## Appendix G

**Table A5.** Rainfall events and main characteristics during 1 June 2019 to 30 June 2019.

| Rainfall Event | Rainfall Intensity/mm/h | Rainfall Duration/h | Maximum 1 h Rainfall Intensity/mm/h | Remarks |
|---|---|---|---|---|
| 15 June 2019 | 0.6 | 16.2 | 3.6 | |
| 22 June 2019 | 0.7 | 33.7 | 6.6 | |
| 25 June 2019 | 1.8 | 6.0 | 3.6 | |
| 3 July 2019 | 0.9 | 19.8 | 2.6 | |
| 8 July 2019 | 1.5 | 9.2 | 3.6 | |
| 12 July 2019 | 0.8 | 13.5 | 5.4 | No runoff data |
| 15 July 2019 | 0.7 | 5.8 | 3.0 | appeared |
| 18 July 2019 | 1.4 | 11.8 | 4.8 | |
| 6 September 2019 | 3.7 | 2.7 | 6.6 | |
| 8 September 2019 | 0.8 | 11.8 | 5.8 | |
| 9 September 2019 | 0.9 | 4.3 | 0.8 | |
| 18 September 2019 | 0.8 | 22.7 | 2.6 | |
| 19 July 2019 | 4.6 | 2.8 | 7.2 | Runoff data appeared but no obvious peak flow |
| 22 July 2019 | 11.2 | 0.8 | 9.0 | |
| 6 August 2019 | 1.4 | 19.2 | 5.2 | Runoff data appeared, but there were no significant differences of the peak flow among each runoff plot because of the influence of previous rainfall events |
| 8 August 2019 | 29.9 | 1.8 | 52.2 | |
| 9 August 2019 | 7.3 | 2.5 | 14.7 | |
| 9 June 2019 | 2.0 | 17.0 | 18.4 | Typical long-duration moderate rainfall |
| 28 June 2019 | 2.5 | 22.2 | 24.8 | Typical long-duration rainstorm |
| 4 August 2019 | 30.0 | 1.7 | 49.4 | Typical short-duration rainstorm |

## Appendix H

**Table A6.** Correlation between peak flow and the composite indexes of the spatial pattern of cypress (*Cupressus funebris*) and micro-topography.

| Peak Flow | Composite Index of Topography | Composite Index of the Spatial Pattern of Cypress (*Cupressus funebris*) |
|---|---|---|
| Peak flow(Long-duration moderate rainfall) | 0.929 ** | −0.758 * |
| Peak flow(Long-duration rainstorm) | 0.857 ** | −0.816 ** |

* denotes significant differences at $p < 0.05$ level, and ** denotes significant differences at $p < 0.01$.

## Appendix I

**Table A7.** Characteristics of the underlay in different regions of the response surface.

| Response Surface Area | Rainfall Conditions | Composite Index of Topography | Composite Index of the Spatial Pattern of Cypress | Topographic Relief/m | Surface Roughness | Runoff Path Density/(m/100 m²) | Contagion Index of Cypress | Stand Density of Cypress/(Plants/100 m²) |
|---|---|---|---|---|---|---|---|---|
| I | Long-duration moderate rainfall | 5.5–10.5 | 10–22.5 | 1.51–1.54 | 1.65–1.72 | 7.9–8.02 | 0.4–0.43 | 34–36 |
| | Long-duration rainstorm | 5.5–12.5 | 10–20.5 | 1.36–1.71 | 1.21–1.72 | 7.9–11.22 | 0.4–0.53 | 22–36 |
| II | Long-duration moderate rainfall | 7.5–12.5 | 10–31 | 1.36–1.71 | 1.21–1.72 | 8.02–11.22 | 0.42–0.53 | 22–36 |
| | Long-duration rainstorm | 9–12.5 | 10–41 | 1.36–1.71 | 1.21–1.35 | 9.62–11.22 | 0.42–0.53 | 22–36 |
| III | Long-duration moderate rainfall | 5.5–7.5 | 18–41 | 1.31–1.56 | 1.45–1.62 | 6.96–7.52 | 0.47–0.64 | 54–70 |
| | Long-duration rainstorm | 5.5–9 | 15–41 | 1.31–1.56 | 1.45–1.73 | 6.96–9.92 | 0.4–0.64 | 34–70 |
| IV | Long-duration moderate rainfall | 5.5–12.5 | 22.5–41 | 1.31–1.56 | 1.45–1.73 | 6.96–9.92 | 0.47–0.64 | 54–70 |
| | Long-duration rainstorm | 5.5–12.5 | 20.5–41 | 1.31–1.56 | 1.45–1.73 | 6.96–9.92 | 0.47–0.64 | 54–70 |

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
