# Peer review of "Effects of Underlay on Hill-Slope Surface Runoff Process of Cupressus funebris Endl. Plantations in Southwestern China"

_forests, doi:10.3390/f12050644_

Round 1
Reviewer 1 Report
The authors provide an interesting research topic. I have mentioned several general comments in the manuscript, however, they are essential.
The introduction missed a paragraph about hillslope hydrology (surface flow, near-surface flow). I like the 10 plot-design and I can imagine that research works in such a steep terrain can be very hard.
The analysis needs improvement and maybe more measurements. There is a lack of soil properties information. In general, surface flow is created according to topography, vegetation cover or soil properties (infiltration capacity, porosity, ..). Moreover, it is important that variables need to be measured in the field as much as possible. The authors used several indexes to describe the study area (which is fine), however, using too many indexes in the evaluation is confusing. Incorrect conclusions came from the fact that many of the chosen variables (such as surface roughness or runoff path density) can be a product of surface flow (erosion) - not a reason for them.
Therefore, I would suggest revising the manuscript and add more important measured parameters which cause or effect surface flow directly.

Author Response
- Introduction:What about the role of a soil cover in surface flow topic?
Response: According to the Reviewer’s suggestion, we added references to the impact of soil on surface runoff. Details are shown in Line 49-51.
- Line 76,water convergence,What does it mean?
Response: According to the Reviewer’s suggestion, we changed the relevant expression. Details are shown in Line 92-93.
- Line 89,Add more literature about surface and subsurface flow, or in general, hillslope hydrology.
Response: According to the Reviewer’s suggestion, we have supplemented the relevant literature to clarify our research goals. Details are shown in Line 111-119.
- Line 102,This is insufficient description of soil. What about infiltration capacity, porosity or other soil parameters which are responsible for surface flow? How deep is the organic litter? Is there variability in the infiltration capacity. Is this soil prone to erosion?
Response: According to the Reviewer’s suggestion, we have added related indicators such as soil physical properties, infiltration, etc. It should be noted that the soil samples are collected before the construction of the runoff plots. If the soil samples are collected after the establishment of the plots, it will cause man-made damage to the terrain conditions of the community. Details are shown in Line 127-137.
- Line 148,It is not clear why you refer to this article. What methods do you use?
Response: According to the Reviewer’s suggestion, we have changed the relevant references. Details are shown in Line 216 and 780.
- Line 159,better "estimated"
Response: According to the Reviewer’s suggestion, we changed the expression. Details are shown in Line 227.
- Line 206,I am not sure if I understand this paragraph correctly. The problem is that vegetation affects soil and soil together with rainfall causes surface flow.
Response: According to the Reviewer’s suggestion, we have changed the relevant expression. Details are shown in Line 282-283.
- Line 207,the factors of the spatial patterns, Which?
Response: According to the Reviewer’s suggestion, we have added related expressions. Details are shown in Line 285-287.
- Line 214,Does not make sence. There is to much bias inside. What does mean topographic reliaf and how did you calculate it? What is the value of this equation?
Response: According to the Reviewer’s suggestion, we explained the meaning of the formula. Details are shown in Line 295-304.
- Line 219,characteristic parameters of the runoff process,which are they?
Response: According to the Reviewer’s suggestion, we have added related expressions. Details are shown in Line 288.
.
- Line 238,surface vegetation coverage,How do you know? How did you measure this?
Response: We deleted the first two paragraphs of 3.2 and moved the related content to 2.2.1. As for the surface vegetation coverage, we have added relevant expressions. Details are shown in Line 161-163.
- Line 246,canopy interception,What about grass interception
Response: According to the Reviewer’s suggestion, we have added relevant expressions. Details are shown in Line 168.
- Line 248,This is not the only reason. What about the role of organic litter, the surface organic layer on soil? Is there a difference in organic soil layer thickness?
Response: It may be that our expression is not clear, so the structure of the article has been adjusted. The purpose of this study is to clarify the variables that cause the differences in runoff among various runoff plots, and does not deny the impact of surface vegetation and soil on runoff. We have added relevant expressions. Details are shown in Line 165-174.
- Line 255,This is already know that these factors are connected with runoff formation in general but how do you know they are the key factors? For example, runoff path density is the output of soil and vegetation properties and mainly of water. The r. p. density is affected by surface flow, not vice versa.
Response: According to the Reviewer’s suggestion, we have changed the relevant expression. Details are shown in Line 357-361.
- Line 409,As mentioned above, this is only partly true.
Response: According to the Reviewer’s suggestion, we have changed the relevant expression. Details are shown in Line 620-623.
- Line 414,The blocking effect was not tested this manuscript.
Response: It may be that the appendix is not placed in the main text. We use the peak flow coefficient to reflect the blocking effect of the underlay. Details are shown in Line 334.
- The introduction missed a paragraph about hillslope hydrology (surface flow, near-surface flow). I like the 10 plot-design and I can imagine that research works in such a steep terrain can be very hard.
The analysis needs improvement and maybe more measurements. There is a lack of soil properties information. In general, surface flow is created according to topography, vegetation cover or soil properties (infiltration capacity, porosity, ..). Moreover, it is important that variables need to be measured in the field as much as possible. The authors used several indexes to describe the study area (which is fine), however, using too many indexes in the evaluation is confusing. Incorrect conclusions came from the fact that many of the chosen variables (such as surface roughness or runoff path density) can be a product of surface flow (erosion) - not a reason for them.
Response: According to the Reviewer’s suggestion, we have adjusted the structure of the full text and supplemented the relevant content mentioned. In the Results and analysis section, we have added quantitative analysis. In the Discussion section, we have added the vegetation pattern adjustment strategy.

Reviewer 2 Report
“Effects of underlay on hill-slope surface runoff process of Cupressus funebris plantations in southwestern China”
Something “surprising”, I found on the literature (ResearchGate) two references from the authors with similar results.
Authorea looks a « document editor” cf. https://www.authorea.com/product
Bingchen Wu, Shi Qi. Coupling effects of topography and the spatial distribution of cypress on surface runoff coefficient on a steep forested slope in southwest China. Authorea. September 11, 2020.
DOI: 10.22541/au.159986506.61880487
Bingchen Wu, Shi Qi. Effects of underlay on hill-slope surface runoff process under different rainfall intensities. Authorea. November 13, 2020. DOI: 10.22541/au.160526725.57027526/v1
If we disregard these 2 documents, this study shows interesting results especially for runoff processes on steep slopes in mountainous areas of southwestern China. At regional scale, the results are interesting but in term of research since long time scientist provide high correlation between heavy rains, vegetal cover, and soil surface features.
Weak points: some information about soil surface features (crust, bulk density) and runoff and rain characteristics are unclear or missing (herbaceous + litter layer(s)).
Strong point: the specific spatial pattern of Cupressus funebris is clearly defined.
This paper should be improved specifically for the definition of some studied parameters (type of rain, micro-habitats, peak flow) and the presentation of the results. This is the first step to understand your discussion.
The document should be improved following the comments below:
- Introduction
Line 60: what is the definition of the “micro-habitats”?
- Materials and methods
Do you have some data about soil texture specially the first layer (0 to 5 or 10 cm)?
At soil surface do you have crusts? Free sands? Gravels?
- Line 175 to 178:
Describe the automatic rain gauge collecting the rain for each plot (Tipping buckets?)
Describe the system to collect the runoff with volume of the water/runoff tank and his equipment (with direct outlet or/and repartition tube? What is happening when the tank is full?)
- Line 182 to 190 – this paragraph is not explicit enough, unclear.
Rain intensity is generally expressed in mm h-1.
So, I am little bit confused with the daily rainfall intensity. Is it not rather the daily rain (high of rainwater in 24h)?
Providing more information about the method to calculate the rainfall characteristics especially how you define i) the long duration moderate rainfall, ii) the long duration rainstorm, and iii) the short duration rainstorm.
Why did you take in account for your survey/experiment only 20 rains for a total of 410.1 mm while indicating that the annual rainfall is 1200 mm?
Figure 2,3,4:
- Why is runoff intensity not expressed in mm h-1? What is the Time(10mn)?
it would be interesting to have these figures completed with the rainfall intensity in 1 hour on a Y’ axis.
- What is the “surface runoff”? The peak flow?
- Could you define what is the peak flow for you?
- Results – complete your explanation concerning some results concerning runoff and peak flow.
Apparently, the microtopography is so important but you do not provide information about soil surface features which determine the runoff and the peak flow specifically for short duration rain event (with high rain intensity I suppose).
What is the volume of total runoff comparing with the rain at least for your 20 events? Runoff coefficient…
Do you have presence of crusting process? Soil compaction? Do you have bulk density data?
Do you have data for shrub, herbaceous cover or/and litter? In the appendix B, I am not sure that the column of vegetal coverage included tree and other vegetal layers such as shrub and herbaceous.
- Discussion – complete your explanation concerning:
Line 250-251-363: what are the micro-habitats?
How the runoff increased the roughness of the soil? The micro-relief is probably modified by the succession of rain events, but the roughness of the soil is also induced by the presence of herbaceous layer, litter, gravel, rocks, runoff, rills, gullies, fauna, etc.
Could you discuss this important point?
- Have a better conclusion and perspectives.
The “blocking effect” of different underlays on surface runoff decreased with the increase of rainfall intensity and the concentration of precipitation. This is not a surprise. You are suggesting that subsequent studies should be done on the rainfall intensity thresholds corresponding to the effect of each factor on the peak flow. Which ones? How?
The conclusion should improve better the take-home message of this article with focus of importance to clarify the impact of underlay on surface runoff ans soil loss on steep slope ecosystems.
- References
You have a lot of references but too many of them are in Chinese = 11 (difficult to collect the papers and few reviewers can check the contains). Be careful the reference #33 is incomplete.
Please to “discuss” about surface runoff process (and peak flow) on steep slope (with roughness, crusting, soil compaction, rills, etc.) depending of rain event characteristics and vegetal covers (type, throughfall, stemflow), at least you could be read or/and add recent paper such as:
Janeau, J.L., Grellier, S., Podwojewski, P., 2015. Influence of rainfall interception by endemic plants versus short cycle crops on water infiltration in high altitude ecosystems of Ecuador. Hydrology Research 46, 1008-1018.
Liu B, Wang D, Fu S, Cao W. 2017. Estimation of Peak Flow Rates for Small Drainage Areas. Water Resources Management 31: 1635-1647. DOI: 10.1007/s11269-017-1604-y
Patin J, Mouche E, Ribolzi O, Sengtahevanghoung O, Latsachak KO, Soulileuth B, Chaplot V, Valentin C. 2018. Effect of land use on interrill erosion in a montane catchment of Northern Laos: An analysis based on a pluri-annual runoff and soil loss database. Journal of Hydrology 563: 480-494.
Wang YS, Cheng CC, Xie Y, Liu BY, Yin SQ, Liu YN, Hao YF. 2017. Increasing trends in rainfall-runoff erosivity in the Source Region of the Three Rivers, 1961-2012. Science of the Total Environment 592: 639-648.
Yu Y, Wei W, Chen LD, Feng TJ, Daryanto S. 2019. Quantifying the effects of precipitation, vegetation, and land preparation techniques on runoff and soil erosion in a Loess watershed of China. Science of the Total Environment 652: 755-764. DOI: 10.1016/j.scitotenv.2018.10.255
Zhang, H.X.;Wu, H.W.; Li, J.; He, B.; Liu, J.F.;Wang, N.; Duan,W.L.; Liao, A.M. Spatial-temporal variability of throughfall in a subtropical deciduous forest from the hilly regions of eastern China. J. Mt. Sci. 2019, 16, 1788–1801.
- Appendix
Appendix A – could you indicate the contain of the “empty” box with 3 dots …
Appendix D – the rain intensity is expressed in mm per 24 h; is it not rather the total daily volume of rain?
We have a confusion between Appendixes DEFH which seems be the Tables 1,2,3,4
Author Response
- Line 60: what is the definition of the “micro-habitats”?
Response: According to the Reviewer’s suggestion, we added the relevant expression. Details are shown in Line 75-76.
- • Materials and methods.
Do you have some data about soil texture specially the first layer (0 to 5 or 10 cm)?
At soil surface do you have crusts? Free sands? Gravels?
Response: According to the Reviewer’s suggestion, we have added related indicators such as soil physical properties, infiltration, etc. Details are shown in Line 128-137.
- Line 175 to 178:
Describe the automatic rain gauge collecting the rain for each plot (Tipping buckets?)
Describe the system to collect the runoff with volume of the water/runoff tank and his equipment (with direct outlet or/and repartition tube? What is happening when the tank is full?)
Response: According to the Reviewer’s suggestion, we added the relevant expression. Details are shown in Line 175-181.
- Line 182 to 190 – this paragraph is not explicit enough, unclear.
Rain intensity is generally expressed in mm h-1.
So, I am little bit confused with the daily rainfall intensity. Is it not rather the daily rain (high of rainwater in 24h)?
Providing more information about the method to calculate the rainfall characteristics especially how you define i) the long duration moderate rainfall, ii) the long duration rainstorm, and iii) the short duration rainstorm.
Why did you take in account for your survey/experiment only 20 rains for a total of 410.1 mm while indicating that the annual rainfall is 1200 mm?
Response: According to the Reviewer’s suggestion, we have changed the relevant expression. The study area has plenty of rainfall, with a lot of light rain throughout the year. Details are shown in Line 247-266.
- Figure 2,3,4:
Why is runoff intensity not expressed in mm h-1? What is the Time(10mn)?
it would be interesting to have these figures completed with the rainfall intensity in 1 hour on a Y’ axis.
What is the “surface runoff”? The peak flow?
Could you define what is the peak flow for you?
Response: According to the Reviewer’s suggestion, we have changed the figure. Details are shown in Line 329-333.The explain of peak flow was shown in Line 256-257.
- Results – complete your explanation concerning some results concerning runoff and peak flow.
Apparently, the microtopography is so important but you do not provide information about soil surface features which determine the runoff and the peak flow specifically for short duration rain event (with high rain intensity I suppose).
What is the volume of total runoff comparing with the rain at least for your 20 events? Runoff coefficient…
Do you have presence of crusting process? Soil compaction? Do you have bulk density data?
Do you have data for shrub, herbaceous cover or/and litter? In the appendix B, I am not sure that the column of vegetal coverage included tree and other vegetal layers such as shrub and herbaceous.
Response: According to the Reviewer’s suggestion, we have supplemented the relevant information. Details are shown in Line 128-137. The purpose of this study is to clarify the difference in peak flow between different runoff plots caused by key indicators in the underlay, and the measured values of these key indicators are stable after long-term evolution. It does not deny the influence of soil infiltration, crusting and other processes on the runoff process, but these factors can be considered as irrelevant variables.
- Discussion – complete your explanation concerning:
Line 250-251-363: what are the micro-habitats?
How the runoff increased the roughness of the soil? The micro-relief is probably modified by the succession of rain events, but the roughness of the soil is also induced by the presence of herbaceous layer, litter, gravel, rocks, runoff, rills, gullies, fauna, etc.
Could you discuss this important point?
Response: According to the Reviewer’s suggestion, we have supplemented the relevant expression about roughness. Details are shown in Line 503-511.
- Have a better conclusion and perspectives.
The “blocking effect” of different underlays on surface runoff decreased with the increase of rainfall intensity and the concentration of precipitation. This is not a surprise. You are suggesting that subsequent studies should be done on the rainfall intensity thresholds corresponding to the effect of each factor on the peak flow. Which ones? How?
The conclusion should improve better the take-home message of this article with focus of importance to clarify the impact of underlay on surface runoff ans soil loss on steep slope ecosystems.
Response: According to the Reviewer’s suggestion, we have added related expressions. Details are shown in Line 624-636.
- References
You have a lot of references but too many of them are in Chinese = 11 (difficult to collect the papers and few reviewers can check the contains). Be careful the reference #33 is incomplete.
Please to “discuss” about surface runoff process (and peak flow) on steep slope (with roughness, crusting, soil compaction, rills, etc.) depending of rain event characteristics and vegetal covers (type, throughfall, stemflow), at least you could be read or/and add recent paper such as:
Response: According to the Reviewer’s suggestion, we deleted some Chinese documents and added the mentioned documents..
Janeau, J.L., Grellier, S., Podwojewski, P., 2015. Influence of rainfall interception by endemic plants versus short cycle crops on water infiltration in high altitude ecosystems of Ecuador. Hydrology Research 46, 1008-1018. (Line 796)
Liu B, Wang D, Fu S, Cao W. 2017. Estimation of Peak Flow Rates for Small Drainage Areas. Water Resources Management 31: 1635-1647. DOI: 10.1007/s11269-017-1604-y. (Line 826)
Patin J, Mouche E, Ribolzi O, Sengtahevanghoung O, Latsachak KO, Soulileuth B, Chaplot V, Valentin C. 2018. Effect of land use on interrill erosion in a montane catchment of Northern Laos: An analysis based on a pluri-annual runoff and soil loss database. Journal of Hydrology 563: 480-494. . (Line 832)
Wang YS, Cheng CC, Xie Y, Liu BY, Yin SQ, Liu YN, Hao YF. 2017. Increasing trends in rainfall-runoff erosivity in the Source Region of the Three Rivers, 1961-2012. Science of the Total Environment 592: 639-648. . (Line 844)
Yu Y, Wei W, Chen LD, Feng TJ, Daryanto S. 2019. Quantifying the effects of precipitation, vegetation, and land preparation techniques on runoff and soil erosion in a Loess watershed of China. Science of the Total Environment 652: 755-764. DOI: 10.1016/j.scitotenv.2018.10.255. (Line 847)
Zhang, H.X.;Wu, H.W.; Li, J.; He, B.; Liu, J.F.;Wang, N.; Duan,W.L.; Liao, A.M. Spatial-temporal variability of throughfall in a subtropical deciduous forest from the hilly regions of eastern China. J. Mt. Sci. 2019, 16, 1788–1801. . (Line 798)
- Appendix
Appendix A – could you indicate the contain of the “empty” box with 3 dots …
Appendix D – the rain intensity is expressed in mm per 24 h; is it not rather the total daily volume of rain?
We have a confusion between Appendixes DEFH which seems be the Tables 1,2,3,4
Response: According to the Reviewer’s suggestion, we have deleted the technical route because we added it at the beginning mainly to facilitate reviewers to understand the intent of the article. We changed the expression of rainfall intensity. Details are shown in Line 682.

Round 2
Reviewer 1 Report
Although there are several sections for the adjustments in the text, the authors have shown great efforts to improve the quality of the article.
Reviewer 2 Report
Although some questions could still be probed, I appreciated the strong effort done by the author to improve the quality of the document (substantial upgrade to clarify the understanding of the text).